# Design, Modeling, and Analysis of Piezoelectric-Actuated Device for Blood Sampling

**Rakesh Kumar Haldkar [1,*], Vijay Kumar Gupta [2], Tanuja Sheorey [2] and Ivan A. Parinov [1]**

[1]   I. I. Vorovich Mathematics, Mechanics and Computer Sciences Institute, Southern Federal University, 344090 Rostov-on-Don, Russia; parinov_ia@mail.ru

[2]   Mechanical Engineering Discipline, PDPM Indian Institute of Information Technology, Design and Manufacturing, Jabalpur 482005, India; vkgupta@iiitdmj.ac.in (V.K.G.); tanush@iiitdmj.ac.in (T.S.)

**\***   Correspondence: rakeshhaldkar@gmail.com; Tel.: +7-989-626-40-88

**Abstract:** In recent years, micro electro-mechanical system (MEMS)-based biomedical devices have been investigated by various researchers for biomedicine, disease diagnosis, and liquid drug delivery. The micropump based devices are of considerable significance for accurate drug delivery and disease diagnosis. In the present study, design aspects of the piezoelectric actuated micropump used for extraction of blood sample are presented. A pentagonal microneedle, which is an integral part of the micropump, was used to extract the blood volume. The blood was then delivered to the biosensor, located in the pump chamber, for diagnosis. The purpose of such low-powered devices is to get sufficient blood volume for the diagnostic purpose at the biosensor located within the pump chamber, with a minimum time of actuation, which will eventually cause less pain. ANSYS® simulations were performed on four quarter piezoelectric bimorph actuator (FQPB) at 2.5 volts. The modal and harmonic analysis were carried out with various load conditions for FQPB. The extended microneedle lengths inside the pump chamber showed improved flow characteristics. Enhanced volume flow rate of 1.256 μL/s was obtained at 22,000 Hz applied frequency at the biosensor location.

**Keywords:** numerical modeling; piezoelectric bimorph actuator; microneedle; diagnostic device; FSI

## 1. Introduction and Purpose of the Study

Blood diagnostics has become a common requirement for investigating the number of diseases, namely, diabetes, HIV, malaria, dengue, thyroid, etc. According to the International Diabetes Federation (IDF), it is now a disease that affects 463 million people worldwide, out of which 187 million do not even know they have the disease. Researchers estimate that the diabetes dilemma will only increase by 2045, with around 700 million people having the disease. There are 180 million patients needing regular diagnostic worldwide, requiring self-diagnostic tools, i.e., a glucometer [1]. The present study was an attempt to design a wearable device to fill the gap.

In recent years, various researchers for diagnosis of disease have investigated piezoelectric-actuated devices. The review on piezoelectric micropump carried out by various researchers suggested their effectiveness. The precision driving control and high performance can be used in the biomedical applications [2–4]. The investigation was carried out on the performance of a valveless micropump with the help of simulation and experimentation. For the simulation, ANSYS and ANSYS CFX were used for the structural and fluid domains, respectively [5]. The numerical analysis was carried out with consideration of the three-way coupling of a piezoelectrically actuated valveless micropump (PVAM) [6].

A piezoelectric-based micropump with cantilever valve was used for drug delivery application [7]. The authors concluded that shorter cantilever valves gave a higher flow rate. Giorgio et al. designed the e-Mosquito system for blood glucose measurement [8]. Tsuchiya et al. developed a blood extraction system, working at 20 V, for diabetic patients to enable self-monitoring of blood glucose. The experiments were carried out at 20 V,

which is a high-energy requirement, as per a wearable device [9]. The simulations on the micropump model, based on finite element analysis, were also carried out [10–12]. The micropump optimization was done with the diameter of the pump chamber, pump membrane, and needle diameter, but other parameters were not considered by the authors [13,14]. The numerical study was carried out on the piezoelectric micropump considering various parameters, namely, valve length, angle, applied voltage, and piezoelectric actuator length [15]. The micropump was pumping 0.75 µL/min of water at the back-pressure of about 5 MPa with two active valve-based micropumps actuated by a paraffin actuator [16].

Sayar and Farouk investigated the effect of actuation voltage and frequency on pump diaphragm deflection and flow rate. They computed time-averaged velocity fields to predict the effect of applied voltage and actuation frequency on the flow rates [17]. With the increase of actuation frequency of the actuator, the flow rate increased up to the first natural frequency [13].

The design of actuator is governed by actuation voltage, ranges of operating frequency, maximum flow rate, and back-pressure. Haldkar et al. in their paper proposed a new design of strip piezoelectric bimorph disc actuator (SPBD) towards achieving a more practically feasible solution. They conducted detailed numerical analysis to observe effect on flow parameters and concluded that strip piezoelectric bimorphs joined with silicone diaphragm gives improved performance compared to the circular disc actuator [18]. Applied voltage and static pressure had a linear relationship for a certain applied frequency. When a piezoelectric actuator was excited at the resonance frequency, the best result for the static pressure head was obtained [19]. The driving system of the micropump was improved by adding a negative-biased field to the driving voltage. The negative bias increases the displacement of the diaphragm and the flow rates [20]. A critical review was carried out in the probable use of micropump technologies in the biomedical applications. The authors reported some basic recommendations for designing a micropump such as drug delivery and blood transport [21].

The piezoelectric micropump was designed and simulated on microneedle architecture. Microneedle is optimized for overcoming the skin resistance and producing a high flow rate at low operating voltages and frequencies [22]. The pentagonal-shaped microneedle gave minimum pain at insertion into the skin [23]. Further, based on the shape effect of the microneedle, the flow characteristics inside the piezoelectric bimorph-actuated micropump were studied and analyzed [24].

In this paper, a new design of a piezoelectric-based micropump is presented. Four quarter piezoelectric bimorph (FQPB) actuator was used for micropump actuation. On actuation at the natural frequency, each quarter can vibrate separately and acts as a triangular cantilever beam, giving maximum possible deflection. On the actuation of the FQPB, the sucked volume into the pump chamber was channelized towards the biosensor location, instead of allowing it to spread all over in the pump chamber. For the same, the microneedle was extended inside the pump chamber with a hole (flat edge orifice) towards the biosensor location. The new design blocked the flow path elsewhere and required less volume to be sucked, thereby reducing the time of actuation as well as actuation frequency. The research aimed to design a wrist watch dial-like wearable device, consisting of seven such micropumps. Each micropump would be used for one-time blood sampling. The holding fixture that would hold micropumps would be replaceable so that the next sequence of sampling may be carried out.

## 2. Physical Modeling of the Device

The wearable device mainly consisted of seven micropumps arranged in a wrist watch dial fashion, as shown in Figure 1a. Each micropump consisted of a pump chamber, a pentagonal microneedle, a four quarters' piezoelectric bimorph actuator (FQPB), a silicon diaphragm, and a biosensor, as shown in Figure 1b. The piezoelectric bimorph was made up of two layers of piezoelectric material glued together with the help of epoxy glue. The mechanical properties of the materials are given in Table 1. The working fluid consid-

ered was blood, a non- Newtonian fluid, with a density of 1060 kg/m$^3$ and a viscosity of 0.0035 Pa·s [25].

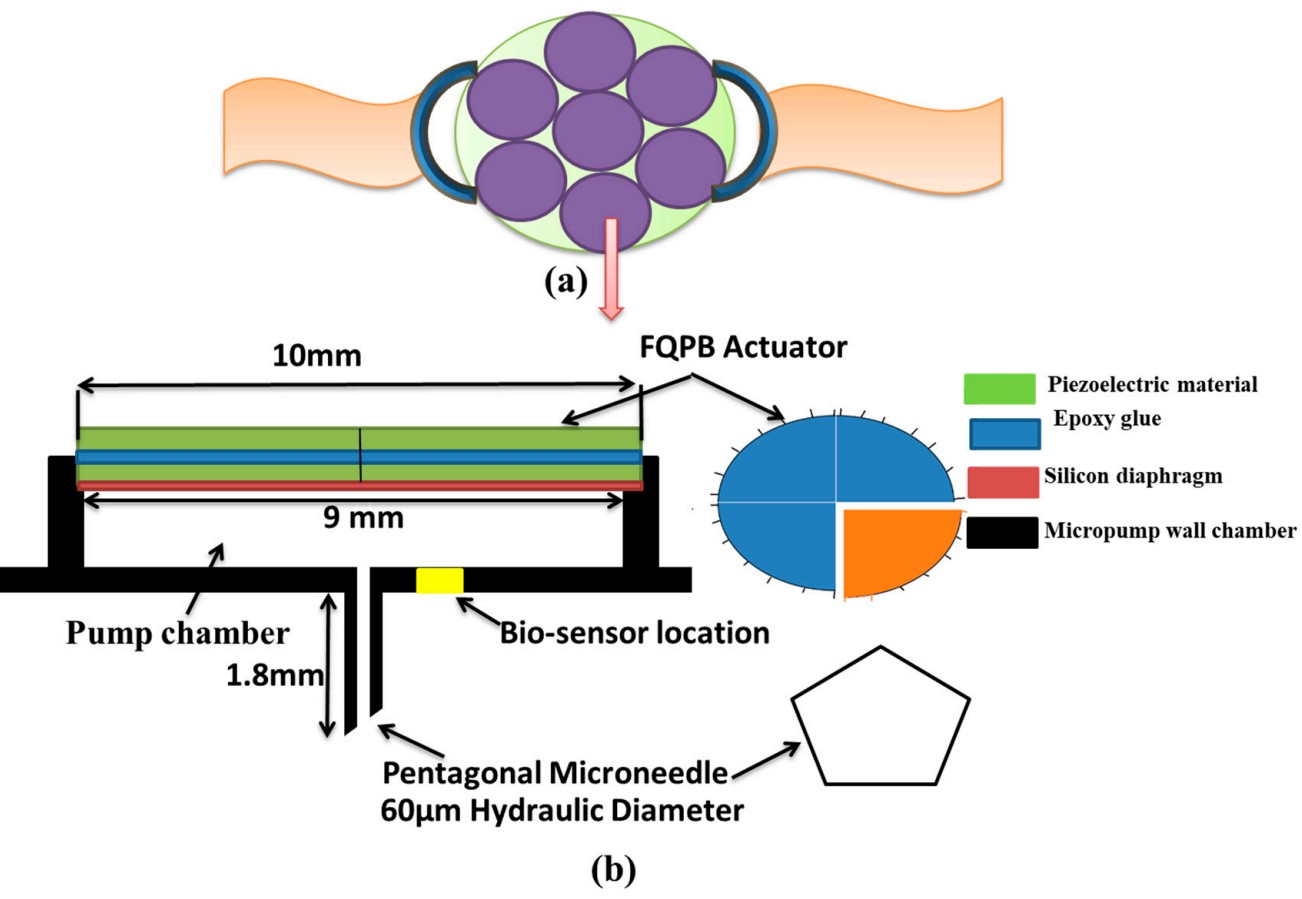

**Figure 1.** Schematic diagram of blood sampling device, (**a**) design of wristwatch with micropumps, and (**b**) detail representation of a piezoelectric micropump with FQPB actuator.

**Table 1.** Mechanical properties of materials.

| Materials | Properties | | Value |
|---|---|---|---|
| Piezoelectric | Piezoelectric charge constant (C/N) | $d_{31} = d_{32}$ | $-320 \times 10^{-12}$ |
| | | $d_{33}$ | $650 \times 10^{-12}$ |
| | Relative dielectric constant | $\varepsilon$ | 3600 |
| | Density (kg/m$^3$) | $\rho$ | 7800 |
| Epoxy Glue | Module of elasticity (GPa) | | 2.478 |
| | Density (kg/m$^3$) | | 1400 |
| Stainless steel | Module of elasticity (GPa) | | 200 |
| | Density (kg/m$^3$) | | 7700 |
| Silicon | Module of elasticity (GPa) | | 168.3 |
| | Density (kg/m$^3$) | | 2329 |

### 2.1. Microneedle

A Microneedle of stainless steel was attached at the bottom of the pump chamber, flush with its inner wall, as shown in Figure 1b. The purpose of the microneedle is to extract the blood from the human body, so as to provide sufficient blood sample for diagnosis at the biosensor location. A regular, pentagonal microneedle of 1.8-mm length having a hydraulic diameter of 60 μm was considered [23,26].

### 2.2. Biosensor

The basic aim of the device under consideration is to analyze the glucose level in the blood sample collected, which is being used for diagnosis of several diseases. A biosensor is located in the pump chamber at a radial distance of 1 mm from the center of the microneedle so that the collected blood can react with the biosensor and glucose level in the blood can be measured. The surface area of biosensor is considered as 0.314 mm$^2$ [27,28].

### 2.3. FQPB Actuator

Four quarters' piezoelectric bimorph was used as an actuator. Each of the quarters of the FQPB works as a cantilever-like triangular beam and, hence, provides larger deflection. On actuation, all the four biomorphs move up and down. As a result, the pump chamber opens to the atmosphere. However, suction is required for blood extraction. Hence, the circular silicon diaphragm was used to cover the pump chamber. For the same, its lower surface was glued to the pump chamber along the circumference. The upper surface of the silicon diaphragm was glued to the lower surfaces of all the four biomorphs, creating a closed interface between the pump chamber and the FQPB.

### 2.4. Working Principle

The micropump is actuated by supplying sinusoidal voltage to FQPB. This will move the piezoelectric bimorph up and down, moving the silicon diaphragm the same way in the pump chamber, creating suction and pressure. During the suction cycle, the diaphragm moves upward, creating back pressure inside the pump chamber, resulting in the suction of blood in the chamber through the microneedle. During the pressure cycle, the diaphragm moves downward, creating pressure inside the pump chamber, causing a part of the extracted volume to move towards the biosensor location. One micropump with the integrated microneedle was used for blood glucose measurement only.

The volume flow rate was calculated based on the diaphragm expansion and contraction in the chamber by using the analytical formulation [29] and computational simulations [30]. A more detailed analytical formulation could be very complex for the micropump flow behavior in non-Newtonian fluid. Thus, in this work, micropump flow was analyzed numerically by using finite element modeling. In the first step, an FE model of the four quarters' piezoelectric bimorph actuator (FQPB) was created. The parametric study was carried out with various load conditions. In the second step, the fluid-structure behavior was considered in the pump chamber. The constitutive equations relating the electric and mechanical fields in piezoelectric media are given by Choi et al. [30]:

$$\{T\} = [C^E]\{S\} - [e]\{E\} \tag{1}$$

$$\{D\} = [e]^T \{S\} + \left[\varepsilon^S\right]\{E\} \tag{2}$$

where $T$ is the stress vector, $C^E$ is the elasticity matrix at the constant electric field, $S$ is the elastic strain vector, $e$ is the piezoelectric stress matrix, $E$ is the electric field intensity vector, and $\varepsilon^S$ is the dielectric matrix at constant mechanical strain. Thus, in the piezoelectric actuator analysis, these coupled piezoelectric equations are solved numerically through the following finite element formulation [31]:

$$M_u \ddot{U} + C_u \dot{U} + K_u U + K_{u\varnothing} \Phi = \text{F} \tag{3}$$

$$K_{u\varnothing}^T U + K_{\varnothing\varnothing} \Phi = Q \tag{4}$$

where $C_u$, $M_u$, and $K_u$ are the damping, the mass matrices, and stiffness matrices, respectively; $Q$ and $F$ are the electrical and mechanical loads, respectively; and $\Phi$ and $U$ are the electrical potentials and the displacement vectors, respectively. The dielectric conductivity and the piezoelectric coupling, are given by $K_{\varnothing\varnothing}$, $K_{u\varnothing}$, respectively. The fluid-flow study

is based on the momentum and continuity equations. It is assumed that blood flow is governed by Navier–Stokes equations [32,33]:

$$\rho \left[ \frac{\partial u}{\partial t} + (u.\nabla)u \right] = -\nabla P + \tau_e \nabla^2 u \tag{5}$$

$$\frac{\partial \rho}{\partial t} + \nabla.(\rho u) = 0 \tag{6}$$

In the above equations, $\rho$ is the density, $u$ is the velocity vector, $\tau_e$ is the viscosity, $t$ is the time, and $P$ is the pressure. The software ANSYS® was used for solving the governing equations numerically using the finite volume method, as implemented in the software.

### 3. Finite Element Modeling of the Micropump

A 3D SOLID226 (20-node tetrahedron) element was used for the piezoelectric material and a SOLID186 (20-node tetrahedron) element was applied for the epoxy glue and silicon diaphragm. A fluid region, named as "Fluid region", was developed in the pump chamber using the FLUID142 element of ANSYS. Refined meshing was used for flow through the microneedle. Material for the pump chamber wall and the microneedle was considered as stainless steel and meshed using the SOLID186 element (20-node tetrahedron element). Figure 2a shows the meshing of the FQPB actuator with the silicon diaphragm. The complete micropump was analyzed for flow characteristics. Meshing of the complete micropump is shown in Figure 2b.

### 3.1. Fluid–Structure Interaction (FSI)

The FSI was used to create interaction between solid and fluid surfaces inside the pump chamber due to the actuation of FQPB. Near the first natural frequency, harmonic voltage was applied to the FQPB. Actuation of FQPB caused pumping action to provide the required flow of blood into the pump chamber due to the combined effect of the pressure and suction cycle.

In ANSYS, solution for fluid and solid regions was carried out by different solvers. CFX fluid solver solved the fluid dynamic problem while multi-physics solid solver solved the structural dynamic problem. Two-way coupling was used to exchange information between the fluid–solid solvers. In one-way coupling, only fluid pressure acting at the structure was transferred to the MPS solver. In two-way-coupling, in addition to the above, the displacement of the piezoelectric bimorph was also transferred to the CFX solver.

### 3.2. Boundary Conditions

For the blood flow analysis, the following boundary conditions were used in the model, as shown in Figure 2c. The meshed area represents the pump chamber and microneedle flow domain.

#### 3.2.1. Opening Condition

The opening boundary condition was applied to the inlet of the microneedle and pressure was considered 100 kPa, equivalent to human blood pressure.

#### 3.2.2. Wall Condition

The peripheral and bottom wall of the pump chamber were solid region and, hence, wall condition was applied.

#### 3.2.3. Symmetry Condition

At the central plane, the symmetry condition was applied. For the flow analysis, half of the domain was used.

### 3.2.4. Interface Condition

FSI condition was applied at the interface of the diaphragm and fluid region. Under this condition, the fluid region followed the shape of the diaphragm.

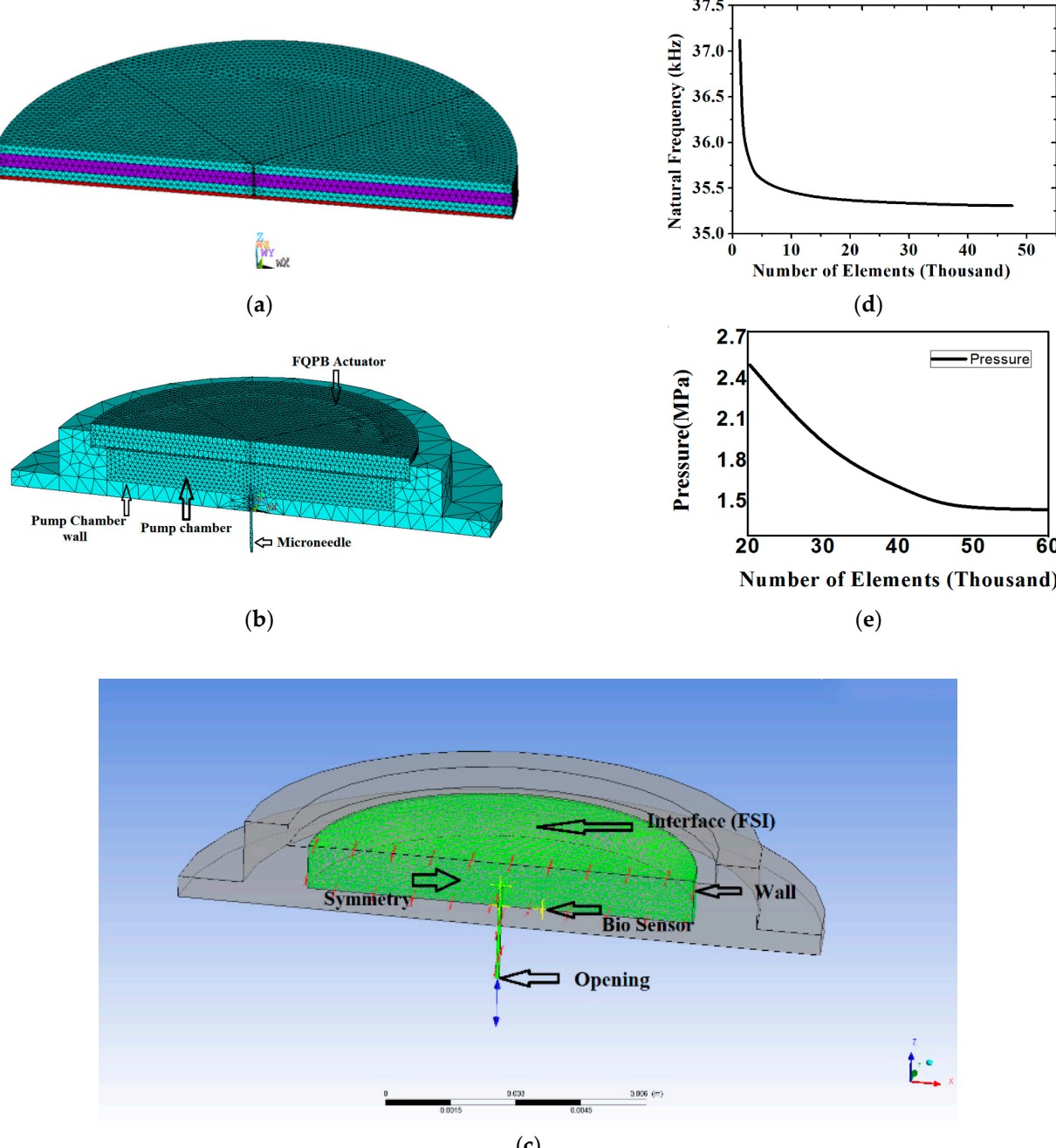

**Figure 2.** (**a**) Meshing of the FQPB actuator with silicon diaphragm, (**b**) meshing of the complete micropump, (**c**) boundary conditions at various interfaces, (**d**) mesh independency test of the actuator, and (**e**) mesh independency test of the complete micropump.

### 3.3. Mesh Independence Test for FQPB and Micropump

A mesh independence test was carried out next. The meshing of FQPB along with the silicon diaphragm and pump chamber was carried out. The meshing of the FQPB actuator varied the number of elements, ranging from 1000 to 50,000. With an increasing number of elements, a change in natural frequency was observed (Figure 2d). It can be seen clearly

that, beyond 28,500 elements, the curve became asymptotic, indicating mesh independence. At 28,500 elements, a natural frequency of 35,338 Hz was observed. An increase in the number of elements beyond 28,500 brought a deviation of the order of 0.09% only, hence, 28,500 elements were considered for the analysis.

A mesh independence test was also carried out for a complete micropump with solid–fluid interaction. For this purpose, the meshing of the fluid domain was done with a varied number of elements, ranging from 20,000 to 60,400. The results were almost the same, with only 0.61% of pressure deviation beyond 49,000 elements, as shown in Figure 2e.

## 4. Results of the Micropump Simulation

Numerical simulation was carried out first with the bimorph actuator and then with complete micropump. The analysis was carried out on half-domain only, owing to the symmetry. Static and modal analyses were carried out to observe the deflection characteristics of the actuator. CFX analysis was carried out to observe flow characteristics of the micropump under various flow conditions. The results are presented in this section.

### 4.1. Static and Modal Analysis of the Actuator (FQPB)

From the literature overview of available devices in the market, meant for the purpose, it was found that such devices work in the range from 2.5 to 5 V, requiring less battery power. With the aim of reducing the actuation voltage, static and modal analyses were carried out with 2.5 V. A deflection of 32.2 nm was observed under static analysis.

### 4.2. Flow Analysis with Flushed Microneedle

The effect of capillarity was analyzed by carrying out numerical simulation. It was found that the capillary force is not sufficient to suck the blood and deliver it to the biosensor location. Therefore, capillary forces were neglected during the simulations.

Simulation of the micropump was carried out with flushed microneedle, applying sinusoidal voltage, as given in Equation (7), to the actuator for studying behavior under harmonic excitation (dynamic mode):

$$V_p = V \sin (2\pi f t) \tag{7}$$

where $V_p$ is the applied voltage, $V$ = 2.5 V, $f$ is the applied frequency (35,338 Hz), and $t$ is the time.

Figure 3a shows flushed microneedle attached to the pump chamber. Streamlines can be seen diverting in all the directions inside the pump chamber (Figure 4a). Location of the biosensor was considered as 1 mm away from the microneedle. The reason for the location is that the extracted blood gets diverted and then aligns with the bottom of the pump chamber at some downstream location from the microneedle outlet as shown. It would ensure sufficient volume of blood reaching the biosensor. Upstream of this location, fluid was not attached to the pump chamber bottom wall. The location of reattachment depends on several variables, namely, pressure difference, actuation voltage, and frequency. The smaller the locational distance, the lesser will be the time of actuation for a particular set of these parameters.

Pfützner et al. [34] reported that a blood volume of 0.6–1.1 µL is required at the biosensor for the diagnosis. Numerical simulation of the base case provided a volume flow rate of 3.4 µL/s (~at 1.2 m/s) at the pump chamber inlet, which is more than three times the required volume. However, only 0.471 µL/s was found to be available at the biosensor location. The reason for the difference in extracted and available volume flow rate is due to the extracted blood getting spread all over in the pump chamber.

### 4.3. Flow Analysis with Modified Microneedle

There are many ways to enhance extracted blood volume. The micropump can be actuated for longer duration or at higher voltage. Both these operations would demand more power. Instead, attention was focused on avoiding spread of the extracted volume.

This can be achieved by channelizing the extracted volume towards the biosensor location. Accordingly, microneedle attachment with the pump chamber was modified by:

1.　Extension of the microneedle into the pump chamber,
2.　Provision of the orifice (hole) towards biosensor location, and
3.　Filleting of microneedle tip.

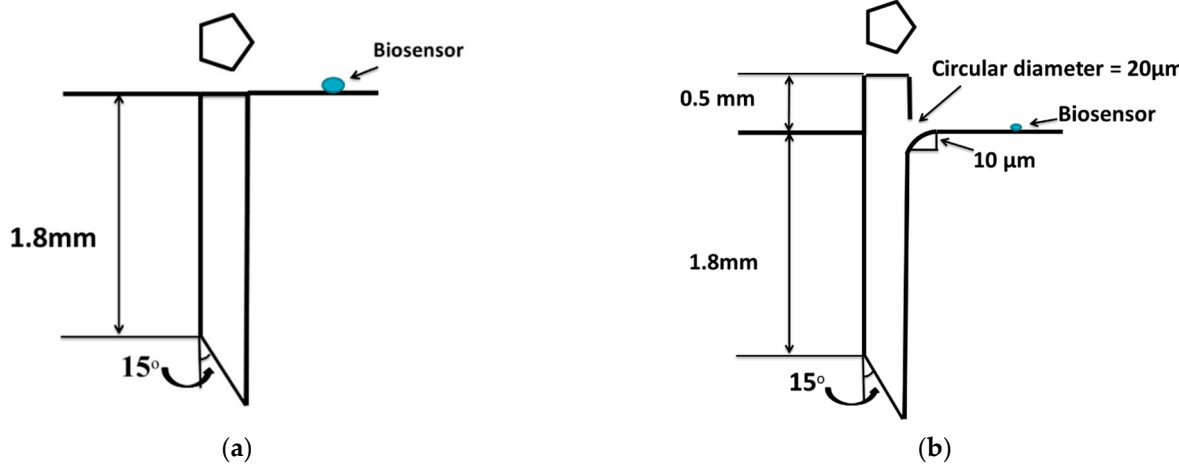

**Figure 3.** Schematic diagram of (**a**) flushed pentagonal microneedle and (**b**) 0.5-mm microneedle extension with a hole.

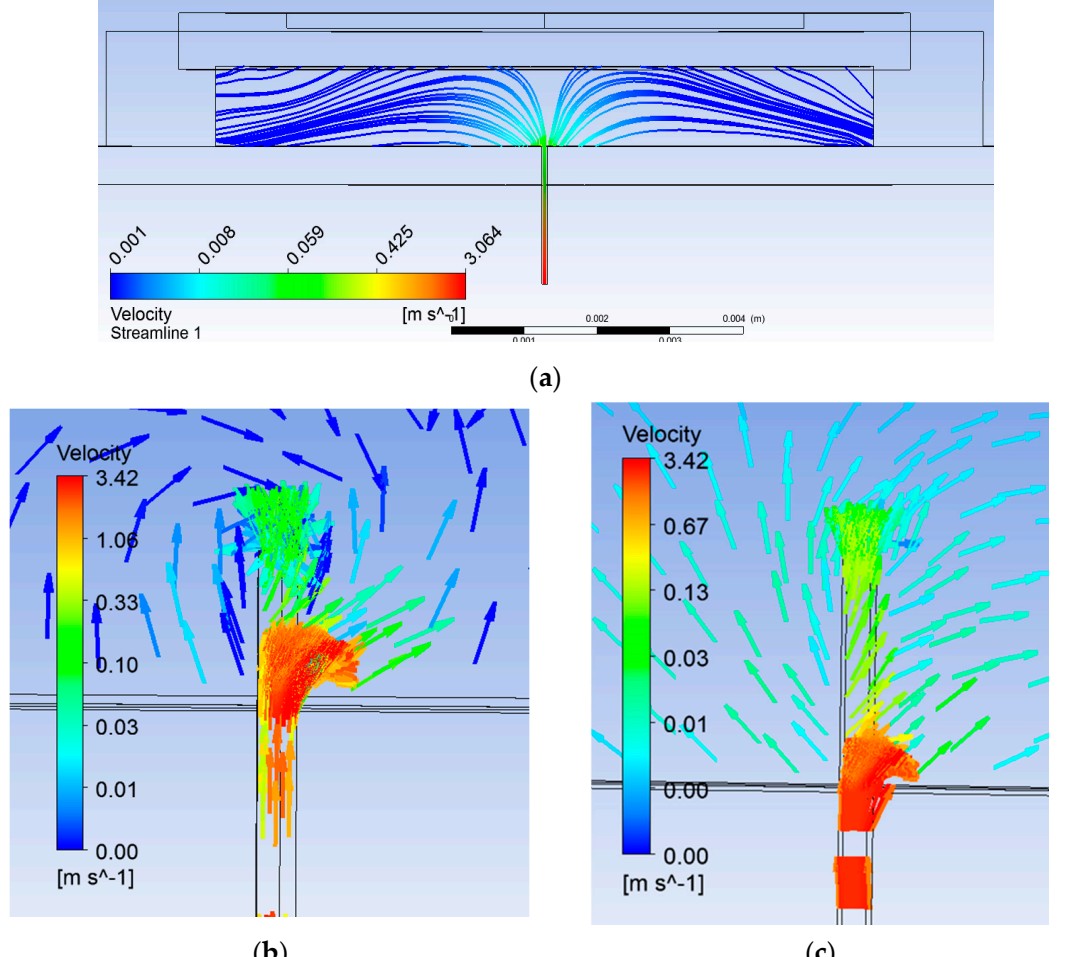

**Figure 4.** Streamline plot (**a**) with flushed microneedle, (**b**) 0.25-mm extended microneedle, and (**c**) 0.50-mm extended microneedle.

Microneedle was modified by extending the microneedle into the pump chamber and providing a hole of 20-μm diameter on the microneedle surface above the pump chamber wall. Additionally, a fillet of a 10-μm radius was provided at the top of the microneedle for a smooth flow of extracted blood from the microneedle to the pump chamber, as detailed in Figure 3b. The fluid, after entering from the microneedle to the pump chamber, was allowed to flow only towards the biosensor, instead of spreading in all directions.

Simulations were carried out with two extended lengths (0.25 mm and 0.5 mm, respectively). Flow characteristics were compared with the flushed microneedle. The selection of extended length was done so as to create appropriate obstruction in the flow path to channelize sufficient volume. With an extended length of the microneedle, flow lines can be seen directing towards the biosensor location through the hole provided for the purpose, as shown in Figure 4b,c. Table 2 summarizes velocity and pressure in all three cases. Flow velocity was observed to enhance by 1.5 and 1.86 times with 0.25-mm and 0.5-mm extended length of the microneedle, respectively.

**Table 2.** Summary of pressure and velocity in the micropump chamber.

| Microneedle Type | Velocity at Biosensor Location (mm/s) | Volume Flow Rate at Biosensor Location (μL/sec) | Max Pressure (MPa) |
|---|---|---|---|
| Flush type | 3 | 0.471 | 8 |
| 0.25 mm extended | 4.5 | 1.413 | 4 |
| 0.50 mm extended | 5.6 | 1.758 | 3 |

The benefits obtained in terms of flow channelization can be seen clearly in Figure 4b,c. This also reduced built-up pressure in the pump chamber by half and less than half, respectively. In addition to channelizing the flow, it served to avoid stagnation of return flow due to the orifice effect, reducing the chaotic flow at the microneedle–pump interface. The flow in the microneedle was fully developed (parabolic profile) and uniform through the orifice. Then, the flow volume through the orifice and the filleted area together were at least half of the extracted volume through the microneedle as compared to one-seventh in the case of the flushed microneedle. The difference in velocity of the blood diverting due to the orifice and that moving up in the pump chamber with the 0.25-mm extended length of the microneedle can be seen clearly in Figure 4b.

Volume flow rate at the microneedle ($f_m$) and orifice ($f_o$) were calculated as follows:

$$f_m = (\pi/4)\,(2d_o)^2\,(2/3)\,U_m \tag{8}$$

$$f_o = (\pi/4)\,(d_o)^2\,U_o \tag{9}$$

where $d_o$ is the orifice diameter, $U_o$ is the velocity at orifice, and $U_m$ is the maximum velocity

The volume flow rate obtained with the 0.25-mm extension of the microneedle was slightly less than half. To further check the advantage of obstruction in the flow path, extension length was increased to 0.5 mm, resulting in more than half the volume flow rate at the biosensor. Figure 5a shows the plot of flow velocity with time. After a short initialization time, flow was seen accumulating and reached to almost a constant average velocity of 5.6 mm/s at around 1600 μs. The volume flow rate available at the biosensor was 1.758 μL/s. The plot clearly shows that the actuation time required was quite less.

$$f = AU \tag{10}$$

where $f$ is the volume flow rate, $A$ is the biosensor area (0.314 mm$^2$), $U$ is the velocity, and $f = 0.314$ mm$^2 \times 5.6$ mm/s $= 1.758$ μL/s (with the 0.5-mm extended microneedle).

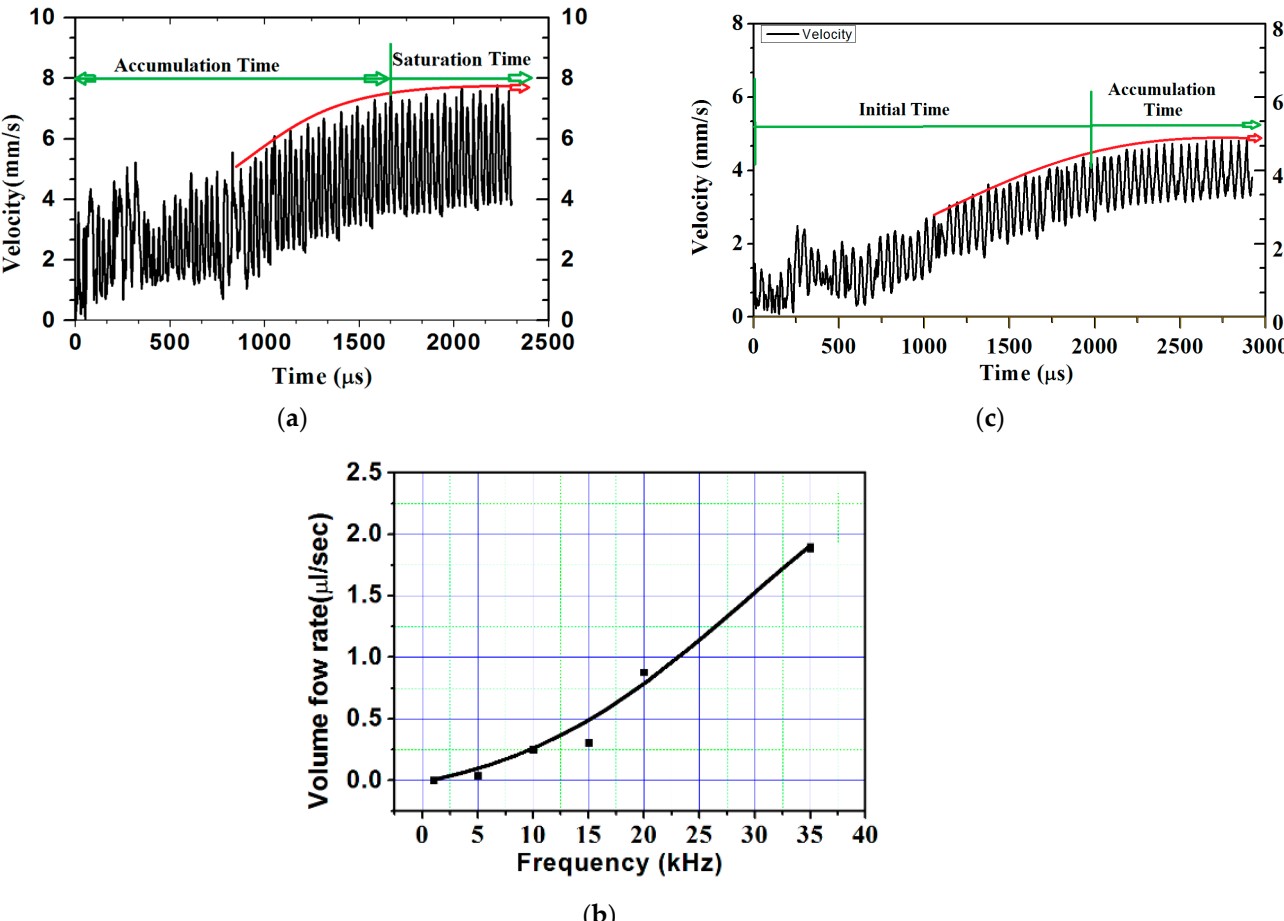

**Figure 5.** Velocity at biosensor location in the case of the 0.50-mm extended microneedle: (**a**) applied frequency of 35,338 Hz, (**b**) volume flow rate vs. applied frequency, and (**c**) applied frequency of 22,000 Hz.

### 4.4. Flow Analysis at Reduced Actuation Frequency

As the volume flow rate available at the biosensor location with the 0.5-mm microneedle extension was 1.6 times higher than the required flow rate, and the actuation frequency was lowered from the first natural frequency. Owing to the direct relation of actuation frequency with pain felt or discomfort, lowering it would further reduce the discomfort due to reduction in suction. Figure 5b shows the variation of the volume flow rate at the biosensor location with actuating frequency. It can be seen from the figure that actuating the FQPB at 22-kHz frequency was able to provide the required blood sample. Hence, further numerical experiments were conducted at actuation frequency of 22 kHz.

Analyses of results obtained for the 0.5-mm extended microneedle at actuation frequency of 22 kHz are presented here. Only 0.90 MPa pressure inside the pump chamber was observed, as compared to 3 MPa at 35.338 kHz frequency, resulting in one-third reduction. This indicated the reduction in pain during the operation of the device. Accumulation started after 2400 μs and average velocity of 4 mm/s was observed, as shown in Figure 5c. From the plot in Figure 6, it can be observed that flow smoothly passed over the biosensor. Based on the same velocity, at the end of 1 s, 1.256 μL of blood was collected inside the pump chamber, which was more than the required volume of 1.1 μL.

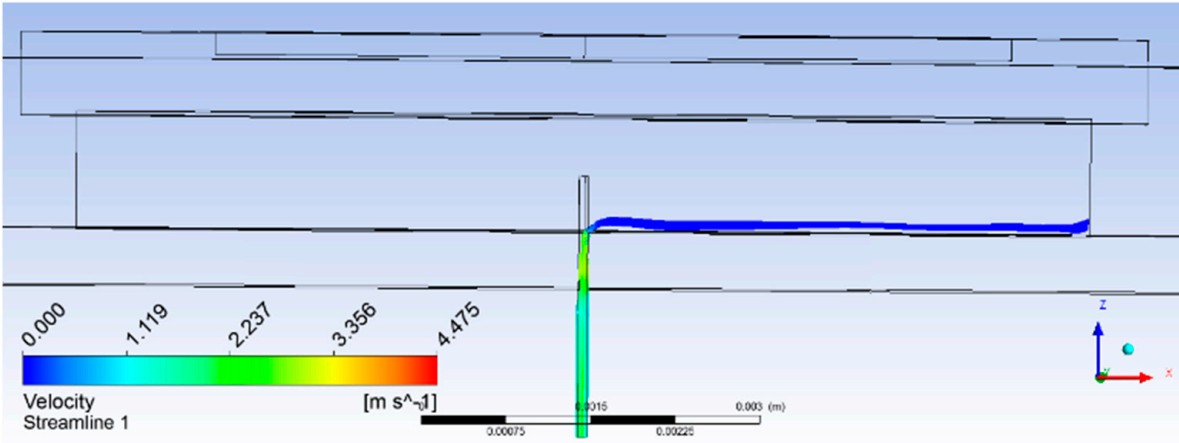

**Figure 6.** Streamlines plot with the 0.50-mm extended microneedle.

### 5. Conclusions

The work aimed to design a wearable device for blood sampling, consisting of seven micropumps to be used seven times for blood sampling. Blood diagnostic requires a definite amount of blood. A micropump should be capable of providing the same. Towards achieving higher deflection, a new design of four quarter piezoelectric bimorph (FQPB) actuator was analyzed. FQPB showed better performance in terms of deflection, effect of reduction in actuation voltage, and frequency. Numerical simulations were carried out on the micropump with the help of ANSYS software. For further improvement in operational parameters, in terms of energy requirement and time of actuation, the micropump was improved to have an extended microneedle length of 0.25 and 0.5 mm inside the chamber, along with the specific directional flow, resulting in enhanced flow. The flow rate of 1.256 µL/s was obtained and compared with the flow rate requirement of blood diagnostic devices to realize the device as a product. The following conclusions were drawn from the present research.

1. The design may be implemented as a wearable device. Considering advancements in the manufacturing of microfluidic systems, manufacturability should not be a problem.
2. The design may be modified to accommodate more sensors for testing other diseases.
3. The FQPB-based micropump with extended microneedle length inside the pump chamber brings the benefit of flow channelization towards the biosensor location, resulting in a reduction in applied voltage (2.5 V) and reduced operating frequency (22 kHz).
4. The simulation results showed that 1.256 µL of the sample was collected at the biosensor in 1 s, which is more than the required volume.
5. The designed device will be easy to use and may allow multiple samplings with a single replacement.
6. The blood was collected in a closed chamber, thereby giving higher accuracy in comparison to the finger-stick, as well as preventing environmental contamination.

**Author Contributions:** R.K.H.: Conceptualization, Methodology, Software, Writing Original Draft, Visualization, Investigation, and Original draft preparation. V.K.G.: Conceptualization, Methodology, Supervision, Software, Validation, Visualization, and Validation of research methods. T.S.: Conceptualization, Methodology, Supervision, Writing, Software, Reviewing, and Editing. I.A.P.: Review, Editing, and Supervision. All authors have read and agreed to the published version of the manuscript.

**Funding:** This research was funded by the grant from the Ministry of Science and Higher Education of Russia supported by Southern Federal University, grant No. VnGr-07/2020-04-IM.

**Institutional Review Board Statement:** Not applicable.

**Informed Consent Statement:** Not applicable.

**Data Availability Statement:** Not applicable.

**Acknowledgments:** The equipment of SFedU and IIITDM Jabalpur India were used. The authors acknowledge the support by Southern Federal University, grant no. VnGr-07/2020-04-IM (Ministry of Science and Higher Education of Russia).

**Conflicts of Interest:** The authors declare no conflict of interest.

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
