# Peer review of "Design, Modeling, and Analysis of Piezoelectric-Actuated Device for Blood Sampling"

_applsci, doi:10.3390/app11188449_

Round 1
Reviewer 1 Report
The authors are working on an important problem in medical diagnostics field, namely the design of wearable diagnostics. The modeling work presented is reasonably clear, however, there are major concerns about the context and application of the results.
The main concern about this work is that it is presented as an innovation in a wearable blood sampling device, however, it is not clear that this device would work or even be manufacturable from the data presented. Overall, I recommend that the authors either demonstrate a prototype or reframe this as a modeling study without strong conclusions about pain perception and with proper caveats as to the manufacutability and use issues. The authors also need to be more clear about what is new in the present study, given their similar previous work (ref 16)
1) A major consideration is whether this device could be made. Even if a prototype is feasible, a manufacturable design may be too costly to fill the need that authors claim to address. A particular concern is how a microneedle would be attached to the rest of the device. Building microneedles up from additive manufacturing processes is very time and resource intensive. Alternatively, attaching a microneedle from another process such as rolled steel seems difficult or impossible with the presented geometries.
2) It is unclear that this design would lead to a wearable sensor that can be used over time. There is no consideration of blood clotting issues, and the modeling is on a time scale that would be most appropriate for a single measurement. If only a single measurement can be done, what is the advantage of a wearable sensor over a finger-stick determination of blood? The authors don't sufficiently address the viscosity and clotting issues in blood (though it is alluded to in one of the references).
3) The relationship to the authors previous work needs to be more clear. Reference 16 shows very similar design elements. For the present work to be published independently, the authors must clarify what is new in the present work.
4) The selection of a pentagonal needle is concerning, as it is not clear that a such a needle could be mass produced. The pentagonal geometry appears to have slower flow at the tip, though it is not clear that this is an advantage. The literature cited to support lower pain sensation is a very limited study and those titanium microneedles were made by a very time-consuming and expensive process.
5) The authors state that "A review paper by Laser et 30 al. emphasized that micropumps suitable for important applications are still not available, 31 and remains a fertile area for future research [2]", however reference 2 is from 2004. Has there been no progress in the last 17 years?
6) I am also interested to know if the authors are aware of the below manuscript, which appears to have a similar design, but is not cited. It would help to know how the present work is different and if/how it builds on this approach.
7) I'm not sure that the conclusion that a reduction in build up pressure causes reduced pain is true. Most pain in microneedle devices is related to the insertion of the device. This conclusion could use further support and explanation.
8) Many microfluidic sampling systems rely on capillary forces. Why is the current approach superior, and should capillary forces be considered in modeling?
Author Response
The authors thank the Reviewer for the careful reading of the manuscript and valuable quary / suggestions. The changes are highlighted with red color.

Reviewer 2 Report
The manuscript discussed numerical simulations of piezoelectric-actuated device for blood sampling. In the manuscript, the authors tried to explain overall numerical procedures of devices by means of commercial software (ANSYS). However, the procedures were acceptable and well known in fields of numerical simulation. Additionally, the manuscript did now provide feature or novelty when compared with previous studies. The following comments are prepared for improving current manuscript substantially.
1. In abstract, motivation, results and discussion were omitted. Please include them.
2. In the section “Introduction”, the authors should show unique or feature of the manuscript while discussing previous studies. Please add and discuss it.
3. In the manuscript, there were so many figures. It would be better to group several figures into one figure. Namely, the figures should be grouped depending on design procedures which authors stressed on. Then, the number of figure will decrease significantly. Additionally, the caption in each figure did not provide sufficient information, even without reading main text. The comment will be applied to several tables.
4. In the section “results and discussion” which was omitted in the manuscript, authors tried to discuss results while compared with previous studies. Finally, the authors stress impact of the current manuscript on the fields of numerical simulation or biomedical engineering.
Author Response

(The authors gave the same response as above.)

Round 2
Reviewer 1 Report
The manuscript is much improved with additional edits and context. I do have concerns as to whether the proposed device can be manufactured, though the modeling is a significant contribution and should be published.
I also suggest integrating the idea of the 7 micropumps into the body of the manuscript. It comes almost as a surprise in the conclusions section and the other important conclusions are overshadowed by introduction of new material.
Author Response

(The authors gave the same response as above.)

Reviewer 2 Report
The manuscript had been improved when compared with the original version. However, the manuscript had been written down as so long, which might lead to avoid reading authors' work. Namely, some parts were still repeated several times. Figures need to be grouped less for improving leadership. For the reason, the revised manuscript should be improved once more for considering publication of the manuscript. The following comments were given for the authors as follows.
- As the reviewer think, the 10 figures might be grouped into four figures or less as below,
(a) Figure 1, 11, and 5 --> Figure 1 : explanation of your goals
(b) Figure 2, 3, and 4 --> Figure 2 : numerical models for ANSYS.
(c) Figure 6 & 7 --> Figure 3 : dynamic responses
(d) Figure 8 * 9 --> Figure 4 : Performance evaluation (volt & frequency)
- In the manuscript, the authors tried to explain or list so many components, which caused to difficulty in reading the manuscript. Thus, it needs to focus on core parts related to yours purposes. That is, as authors proposed a new pump equipped with specific shape, they should provide data for validating their hypothesis appropriately. The reviewer wants to see procedures or results sufficiently related to authors’ goal. However, the contents were spread here and there, without any order. Thus, it needs to trim the revised manuscript. As a simple proposition, core parts remain in the main text. Less important contents will move to supplementary material. Namely, the manuscript should be rearranged for improving the understandability of author’s work.
- As the manuscript is related with numerical simulation, it needs to validate it with experiment data (or results) by reporting previous study or by doing experiments (if authors had prototype of micropump). However, as looking at the figures as results, the reviewer did not find overlapped data of current method or previous method. Namely, instead of parameter studies, authors should try to show improvement or uniqueness of authors’ method while discussing results reported in the previous papers.
Author Response

(The authors gave the same response as above.)

Round 3
Reviewer 2 Report
As the authors tried to reply issues which the reviewer raised appropriately,
the reviewer suggests publication of the manuscript as current form.
Thanks a lot for the authors.